# Target Of Rapamycin pathway in the white-rot fungus *Phanerochaete chrysosporium*

**Duy Vuong Nguyen, Thomas Roret¤, Antonio Fernandez-Gonzalez, Annegret Kohler, Mélanie Morel-Rouhier, Eric Gelhaye, Rodnay Sormani** [ID] *

Université de Lorraine, INRA, IAM, France

¤ Current address: CNRS, LBI2M, Sorbonne Universités, Roscoff, France
* rodnay.sormani@univ-lorraine.fr

**Data Availability Statement:** All relevant data are within the paper and its Supporting Information files.

## Abstract

The Target Of Rapamycin (TOR) signaling pathway is known to regulate growth in response to nutrient availability and stress in eukaryotic cells. In the present study, we have investigated the TOR pathway in the white-rot fungus *Phanerochaete chrysosporium*. Inhibition of TOR activity by rapamycin affects conidia germination and hyphal growth highlighting the conserved mechanism of susceptibility to rapamycin. Interestingly, the secreted protein content is also affected by the rapamycin treatment. Finally, homologs of the components of TOR pathway can be identified in *P. chrysosporium*. Altogether, those results indicate that the TOR pathway of *P. chrysosporium* plays a central role in this fungus.

## Introduction

The Target Of Rapamycin (TOR) signaling pathway is highly conserved among eukaryotes and regulates essential cellular processes including protein synthesis, ribosome biogenesis, autophagy, and cytoskeleton organization [1]. In fungi, the TOR pathway is involved in the response to nutrient resources availability, this being true for carbon and especially sugar and nitrogen [2,3]. TOR is also involved in the stress responses, such as osmotic and oxidative stresses [4–6].

The serine/threonine-protein kinase TOR, firstly identified in *Saccharomyces cerevisiae* [7] is the central component of the TOR signaling pathway. This kinase interacts with other partners to form two multiprotein complexes named TORC1 and TORC2. Both complexes regulate their targets by phosphorylation. Of these two complexes, only TORC1 is sensitive to rapamycin [8,9]. Rapamycin binds to FK506 Binding Protein 12 (FKBP12) and the FK506 Rapamycin Binding (FRB) domain of TOR resulting in the inhibition of the TOR kinase activity and cell growth arrest. The sensitivity to rapamycin has permitted to decipher the TOR signaling pathway in numerous organisms. In fungi, the yeast models *S. cerevisiae* and *Schizosaccharomyces pombe* have been extensively studied even if the first report of the antifungal activity of rapamycin has been obtained with the human pathogenic fungus *Candida albicans* [10]. Since, rapamycin has been successfully tested either with pathogenic fungi such as *Cryptococcus neoformans* [11], *Botrytis cinerea* [12], *Mucor circinelloides* [13], *Fusarium graminearum* [14], or soft rot fungi [15]. Sensitivity to rapamycin has been observed in all fungal

**Funding:** This work was supported by a grant overseen by the French National Research Agency (ANR) as part of the "Investissements d'Avenir" program (ANR-11-LABX-0002-01, Lab of Excellence ARBRE). DVN was supported by a Doctoral Fellowship from the Ministry of Agriculture and Rural Development, Vietnam (Agricultural and Fisheries Biotechnology Program) and support from French ministry of foreign affair (program Campus France). AFG was supported by a postdoctoral grant from Region Grand Est. The funders had no role in study design, data collection and analysis, decision to publish, or preparation of the manuscript.

**Competing interests:** The authors have declared that no competing interests exist.

lineages tested underlying conservation of the targets: TOR complexes and the mechanism of action, the TOR signaling pathway [16].

Most studies on fungal TOR signaling pathways have been conducted in ascomycetes, while data on basidiomycetes other than *C. neoformans* are scarce [13,16]. In *Pleurotus ostreatus* for instance, the *TOR* gene is duplicated and orthologs of the TOR pathway are present suggesting a conservation of that signaling pathway. Recently, a link between SLT2-MAPK, involved in the cell wall integrity signaling, and the TOR pathway has been reported in the edible mushroom and medicinal fungus *Ganoderma lucidum* [17]. In that case, TOR signaling is involved in the regulation of chitin and β-1,3-glucan synthesis and hence of cell wall thickness in a SLT2-MAPK dependent manner [17].

Among basidiomycetes, white rot fungi are very interesting ecological models because of their ability to grow on dead wood and to mineralize it. Wood is considered as a very specific ecological niche: indeed it contains recalcitrant carbon sources, low nitrogen content and potentially a highly toxic environment due to the presence of wood extractives [18]. To adapt to this environment, white rot fungi secrete a large set of enzymes involved in the degradation of wood polymers (cellulose, hemicelluloses and lignin) and possess an extended intracellular detoxification network [19–22]. These two complex machineries require a fine and coordinated regulation system that is up to date largely unknown in particular due to the lack of genetic tools for white-rot basidiomycetes. To test whether the TOR pathway could be involved in regulating the secretion of enzymes involved in lignocellulose decomposition, we tested the effect of rapamycin on *Phanerochaete chrysosporium* growth and secretome composition. By combining genome mining, structure modeling, we identified in this study the TOR signaling pathway of *P. chrysosporium* and highlighted its role in regulating the extracellular secreted protein composition.

## Material and methods

### Fungal strain

The homokaryotic strain *P. chrysosporium* RP78 was used in this study. This is the most widely used strain of *P. chrysosporium* in studies posterior to 2000, especially after its genome sequence was published in 2004 [23]. A version 2.0 of genomic database is available on https://genome.jgi.doe.gov/Phchr2/Phchr2.home.html. The mycelium is maintained on solid malt extract agar medium (20g.L$^{-1}$ and 30g.L$^{-1}$ respectively).

### Growth curve measurement

Germination of *P. chrysosporium* RP78 conidia was measured using a nephelometric reader (NEPHELOstar Galaxy, BMG Labtech, Offenburg, Germany). For inocula, suspensions of spores were obtained from 8 day-old mycelia grown on sporulation medium (Glucose (10 g.L$^{-1}$), malt extract (10 g.L$^{-1}$), peptone from potato (2 g.L$^{-1}$); yeast extract (2 g.L$^{-1}$), asparagine (1 g.L$^{-1}$), $KH_2PO_4$ (2 g.L$^{-1}$), $MgSO_4.7H_2O$ (1 g.L$^{-1}$), thiamine HCl (1 mg.L$^{-1}$), Agar (30 g.L$^{-1}$), all chemicals used were purchased from Sigma Aldrich. Spores were collected with gentle scraping of the agar plates and filtration through Miracloth. The number of spores per 1 mL of suspension was determined with optical density (OD) measurement at 650 nm, and calculated as previously described [24]. For each microplate well, 200 µl of sample were prepared: 10,000 spores were resuspended in 198 µL of malt 1% and 2 µL of rapamycin (LC laboratories, R-5000) were added for treatment or 2 µl of DMSO as mock for control. Equipment was set up with the following parameters: temperature of incubation was 37˚C, cycle time 1 hour and the sum of cycles was 72 hours. Relative nephelometric unit (NRU) values were calculated as previously described [25].

## Sample preparation for experiments on extracellular proteins

$4 \times 10^5$ fungal spores were inoculated in 10 mL Tien & Kirt medium (with and without 1% glucose) and incubated at 37°C with shaking (120 rpm). After 4 days incubation, the fungal biomass was transferred to a new 10 mL of Tien & Kirk medium with 10 μL of DMSO for control, or where 10 μL rapamycin dissolved in DMSO (2 mg.ml$^{-1}$) was added. Samples were then incubated at 37°C with shaking during 48 hours. Supernatants were harvested and stored at -20°C for further measurements. For each condition, 6 replicates were prepared. Fungal biomasses were dried for 48 hours under vacuum at -85°C with a lyophilizer (VirTis BenchTop Pro freeze dryers) and weighed just after drying.

## β-glucosidase activity assay

This enzymatic assay is based on the release of methylumbelliferone (MU) via cleavage of compound MU-β-D-glucopyranoside (MU-G) as the substrate of β-glucosidase (EC3.2.1.3). The stock solutions of the substrate (5 mM) and calibration (25 mM) were prepared in 2-methoxyethanol and were kept at -20°C in the dark. All chemicals were purchased from Sigma Aldrich Chemicals. The working solution of the substrate was prepared by dilution with sterile ultrapure water to the final concentration of 500 μM.

The fluorogenic assay was prepared in a black 96-well microplate. The reaction mixture in each well included 50 μL of sodium acetate buffer (100 mM), 50μl of culture supernatant and 50 μL of working solution of the substrate. The plate also included calibration wells to obtain fluorescene signals for concentration of released MU, which was calculated from the resulting regression line. The plate was then incubated at 25°C and shaken at 600 rpm. 100 μL of stopping buffer (Tris 2.5M, pH 11) was added into mixture of reaction after 15, 30, 45 and 60 minutes of incubation to stop reaction. Measurements were performed in Victor$^3$ microplate reader (Perkin-Elmer life sciences, France) with excitation wavelength set to 360 nm and emission wavelength set to 450 nm.

## Sample preparation and protein quantification

After 20 minutes of centrifugation at 3,000 rpm (model 5810/5810 R, Eppendorf) to remove remaining fungal cells, supernatants from 6 replicates were mixed together and vaporized to remove all liquid under vacuum, at -85°C with a lyophilizer (VirTis BenchTop Pro freeze dryers). Pellets were dissolved with ultrapure sterile water in 0.5 volume of the initial volume. 1 mL of this new supernatant was purified for protein quantification. For purification, proteins were precipitated by mixing 1 mL of secreted sample with 200 μl saturated TCA and incubated overnight at -20°C. Samples were centrifuged at 13,400 rpm at 4°C for 20 minutes. Supernatants were discarded and the pellets were washed twice with cold acetone. The Interchim Protein Quantification BC assay kit was used for protein quantification. Purified proteins were dissolved with 50 μL of 0.2% SDS before adding the working solution of the provider. Reaction mixtures were incubated at 37°C for 30 minutes and at cold room or in a fridge for 5 minutes. To determine protein content, the OD was measured at 562 nm with a spectrophotometer Cary50 (Varian).

## Proteomic analyses

Samples were produced from the supernatants of 6 replicates mixed together for each condition tested. Pellet of 20μg of protein were resuspended in 20 μL Urea, 6 M, Tris, 50 mM, pH 8.0 and samples were processed as follows: Cysteine residues were reduced by addition of 4 mM DTT for 45 minutes, alkylated by addition of iodoacetic acid (IAA) at 40 mM for another 45 min and IAA was blocked by addition of excess DTT (40 mM) for 10 minutes. 180 μL of a solution containing Tris-HCl, 50 mM pH 8.0, CaCl2, 1 mM were added together with 1/100$^{th}$

(weight trypsin/weight protein extract) sequencing grade trypsin and digestion was allowed to occur overnight at 37˚C. Samples were then acidified by addition of 10 μL TFA 10%. Samples were fractionated by NanoHPLC on an Ultimate3000 system equipped with a 20 μL sample loop, a pepMap 100 C18 desalting precolumn and a 15 cm pepMap RSLC C18 fractionation column (all from Dionex). Samples (6.4 μL) were injected using the μlpickup mode and eluted by a 2 to 45% acetonitrile gradient over 60 minutes at 300 nL.min$^{-1}$. Fractions (340, 9 seconds each) were collected with a ProteineerFcII (Bruker) over 51 minutes and eluted fractions were directly mixed on MTP-1536 BC target (Bruker) spots with α-cyano-4-hydroxycinnamic acid (Bruker). LC-MALDI runs dedicated to peptide identification were processed using dedicated automatic methods piloted by WARP-LC software on an Autoflex speed MALDI-TOF/TOF mass spectrometer (Bruker), first in MS mode in the 700–3,500 mass range, using next-neighbour external calibration for all MALDI spots. Thereafter, masses with S/N > 6 were processed for MS/MS analysis in LIFT mode. Peptide assignments were performed from TOF/TOF spectra by Mascot interrogation (Matrix Science) of the *P. chrysosporium* database piloted in Mascot and compiled by Proteinscape with a mass tolerance of 50 ppm in TOF mode and 0.8 Da in TOF/TOF mode, with optional cysteine carbamidomethylation, methionine oxidation and without enzyme cut. The minimal mascot score for peptides was set to 20 and that for proteins was set to 80. Results were cross-validated by interrogating an irrelevant database using the same criteria. Proteins were considered found in a given sample only if identified with a score above 80. When confronted to the random decoy strategy, this score resulted in a false discovery rate < 1%. The other way around, proteins were considered missing only if they were not identified at all. We could not calculate the effective lack of discovery rate in this specific experiment, but in other, technically similar contexts, such proteins exhibited levels at least 20-fold lower than those identified with a score of 80, with less than 2% errors.

## PCR isolation and sequencing of the *PcFkbp12* gene and FRB domains coding sequences

Genomic DNA extractions were performed from 3-day old liquid cultures of WT using QIA-GEN DNeasy plant kit according to the manufacturer's instruction.

To analyze the sequences of *Fkbp12* and of the FRB domain coding sequences, a first step of PCR was done using primers: 5′ACTCAGTCCAACCGTACCTG3′ (forward) and 5′CGAATGACCCGTCGACAATC3′ (reverse) for *Fkbp12* and 5′TCAGTCGAGAGCT CATCAGG3′ (forward) and 5′ACGGCCAACTGAAGATTACG3′ (reverse) to amplify the FRB domain sequences. The PCR reactions were performed using Gotaq Flexi DNA polymerase (Promega) and equipment Mastercycler Nexus41 of Eppendorf. PCR products were visualized on agarose gel then purified using PCR DNA and Gel Band Purification kit (GE Healthcare, UK).

Plasmid pGEMt was used for the ligation of purified PCR products according to manufacturer's protocol (Promega A1360, USA). The reaction mixture includes 5μl of buffer 2X, 1μl of plasmid pGEMt, 3μl of PCR products and 1 μL of T4 DNA ligase. This mixture was incubated overnight at 4˚C and used to transform *Escherichia coli* DH5α. Plasmid purification was performed using PureYield ™ Plasmid Miniprep system kit (Promega) following instructions of the provider. Purified plasmids were stored at -20˚C before being sent to sequencing using T7p and SP6 universal primers (GENEWIZ, UK).

## 3D modeling of the Fkbp12-rapamycin-Tor complex in *P. chrysosporium*

The models of Fkbp12 and the FRB domain of Tor 3D from *P. chrysosporium* were generated by homology modeling. The models used were the FK506-binding protein 1A from *Aspergillus*

*fumigatus* (73% identity; pdb entry:5hwb) for Fkbp12 and the serine/threonine-protein kinase TOR from *human* (63% identity; pdb entry:4jsp) for Tor. Then, the Fkbp12-rapamycin-Tor complex was built by superimposition with the crystal structure of the human complex (pdb entry: 1fap) for Tor1 and Tor2. The side chain and rotamers were optimized using YASARA's refinement protocol [26].

## Results and discussion

### *P. chrysosporium* is sensitive to rapamycin

In the model yeast *S. cerevisiae*, the Tor kinase pathway is the central regulator of the cellular response to nutrient limitation and stresses. Rapamycin treatment that inactivates Tor leads to various effects at different levels and results in cell growth arrest. This prompted us to investigate the effect of rapamycin on *P. chrysosporium* growth. First, various amounts of rapamycin were applied to *P. chrysosporium* spores, and conidia germination rates were analyzed using nephelometric experiments (Fig 1A). In the tested conditions, controls without rapamycin displayed a three-step growth curve, with a lag phase (t = 0h to 7.33h ± 0.5), followed by an exponential phase (t = 7.33h ± 0.5 to 30.9h ± 2.7) and then a stationary phase (after t = 30.9h ± 2.7). The lowest tested concentration of rapamycin (100 ng.mL$^{-1}$) induced an increase both of the lag (37.7h ± 2.5 instead of 7.33h ± 0.5), and exponential (more than 30 hours instead of 20 hours) phases. Higher concentrations of rapamycin (200 and 500 ng.mL$^{-1}$) induced a similar growth pattern, increasing the lag phase and decreasing the measured growth rates during the exponential phase. At 1 μg.mL$^{-1}$ and 2 μg.mL$^{-1}$, rapamycin completely inhibited germination.

The effects of rapamycin were also tested on *P. chrysosporium* growing on solid medium. Three different amounts of rapamycin dissolved in DMSO were dropped on Whatman paper dots placed near *P. chrysosporium* growing mycelia to perform a diffusion test (Fig 1B). During that experiment, growth was observed on control with DMSO but even the lowest amount of rapamycin tested (1 μg) led to growth inhibition (Fig 1B, S1 Movie).

### Effect of TOR inhibition on the protein secretion in *P. chrysosporium*

The secretion of extracellular proteins plays an important role *P. chrysosporium* in for the degradation of lignocellulosic material and toxic compounds. As TOR is known to control various cellular processes through sensing nutritional factors such as sugar availability, the function of PcTor as a regulator of extracellular protein content was evaluated with or without glucose and rapamycin.

A first experiment was performed to obtain secreted proteins in liquid culture. A wild-type strain was grown for 4 days on synthetic medium with 1% glucose. Then, the mycelium was transferred for 48 hours into fresh medium containing 1% or 0% of glucose, with or without 2 μg.mL$^{-1}$ of rapamycin.

Analysis of dry weights of collected mycelia shows an effect of glucose on fungal growth. Rapamycin inhibits growth in both cases, but this effect was more pronounced with glucose (Fig 2A). Quantitatively, the effect of sugar is not visible on extracellular protein content and with or without glucose there is a decrease due to rapamycin treatment in both tested nutritional conditions (Fig 2B). The β-glucosidase activity was also tested on those samples. With glucose, a decrease of β glucosidase activity can be seen (Fig 2C). Glucose deprivation leads to an increase of that activity and rapamycin could counter act that induction (Fig 2C). Taken together, those results indicate that in these experimental conditions sugar deprivation is perceived, a response occurs and rapamycin acts on the three tested parameters.

We used these experimental parameters to assess qualitatively the extracellular protein contents. From the four treatments, the protein pattern of the secretomes on SDS-PAGE gels exhibited differences in presence of rapamycin for both glucose and glucose-deprived

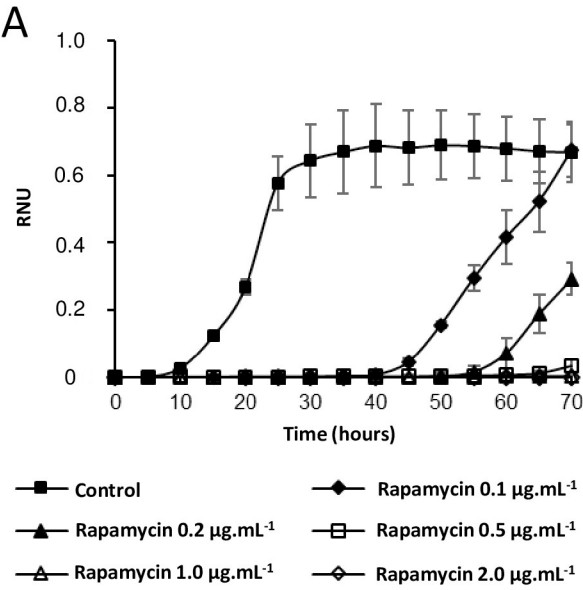

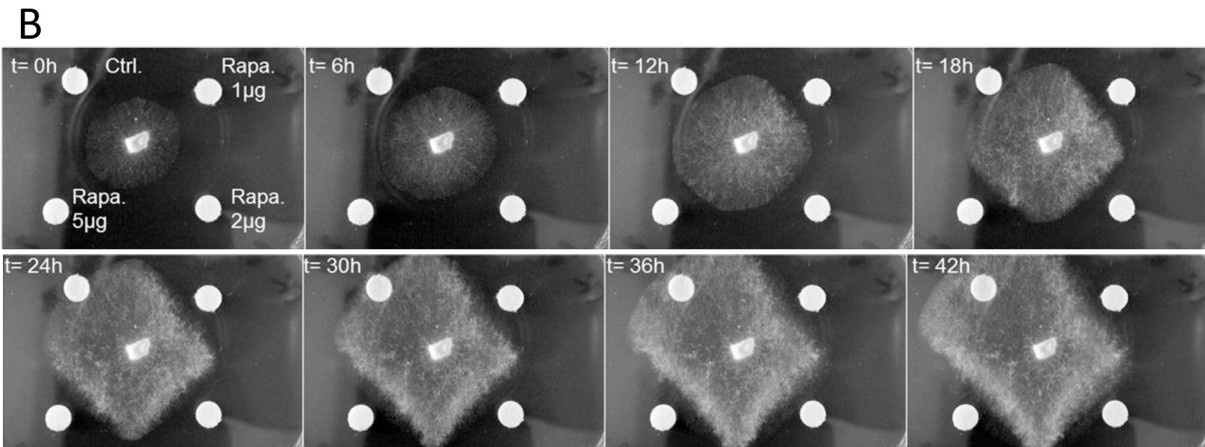

**Fig 1. Effect of rapamycin on *P. chrysosporium* growth.** A. Growth curve measured after germination of conidia from *P. chrysosporium* RP78 in liquid medium using nephelometric measurement with different concentrations of rapamycin. Those curves represent means ± SD of n = 3 replicates. B. Effect of rapamycin on hyphae growth in solid medium (agar plate). Rapamycin dissolved in DMSO was loaded on Whatman filter paper dots, DMSO was used as control, and both were used as in a disk diffusion test. Movie of the corresponding experiment is provided in S1 Movie.

conditions (Fig 2D). Proteomic analyses of the secretomes have been performed for each of the four conditions plus or minus glucose in presence or absence of rapamycin. Samples analyzed were obtained from 6 biological repeats for each treatment. A set of 64 proteins has been identified in those experiments (Fig 2E and S1 Table). These results highlight four points: i. there is a set of 16 proteins found in all four conditions. One of the most abundant proteins found in each case is a copper radical oxidase (AGR57_1123) involved in hydrogen peroxide generation [27] and two GMC oxidoreductases (AGR57_4013 and AGR57_9770). ii. Sugar deprivation increases extracellular protein diversity. Interestingly, four proteins are annotated as protease (AGR57_12626, AGR57_14965, AGR57_1477, AGR57_12049), three are glycoside hydrolases (AGR57_11735, AGR57_12854, AGR57_2053) and an amylase (AGR57_990) can

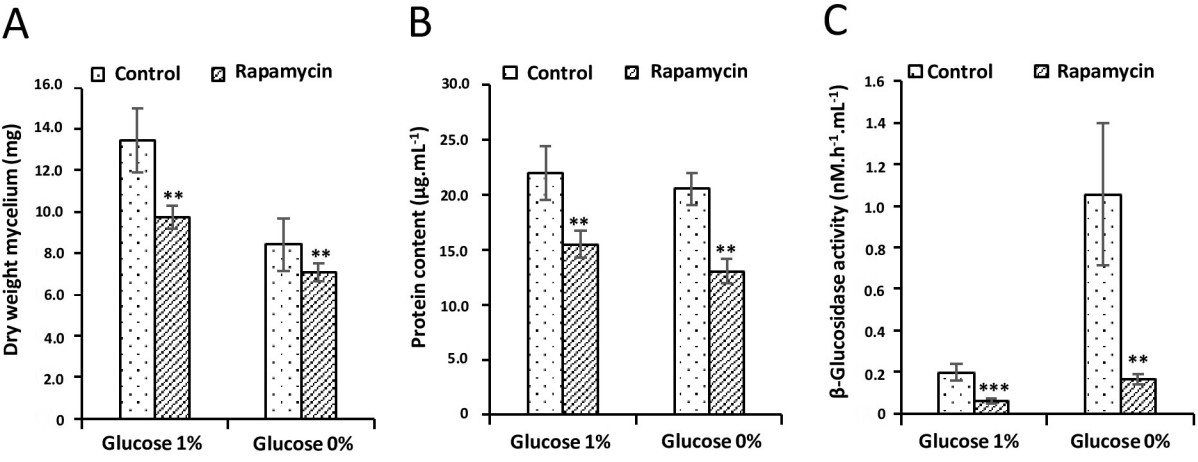

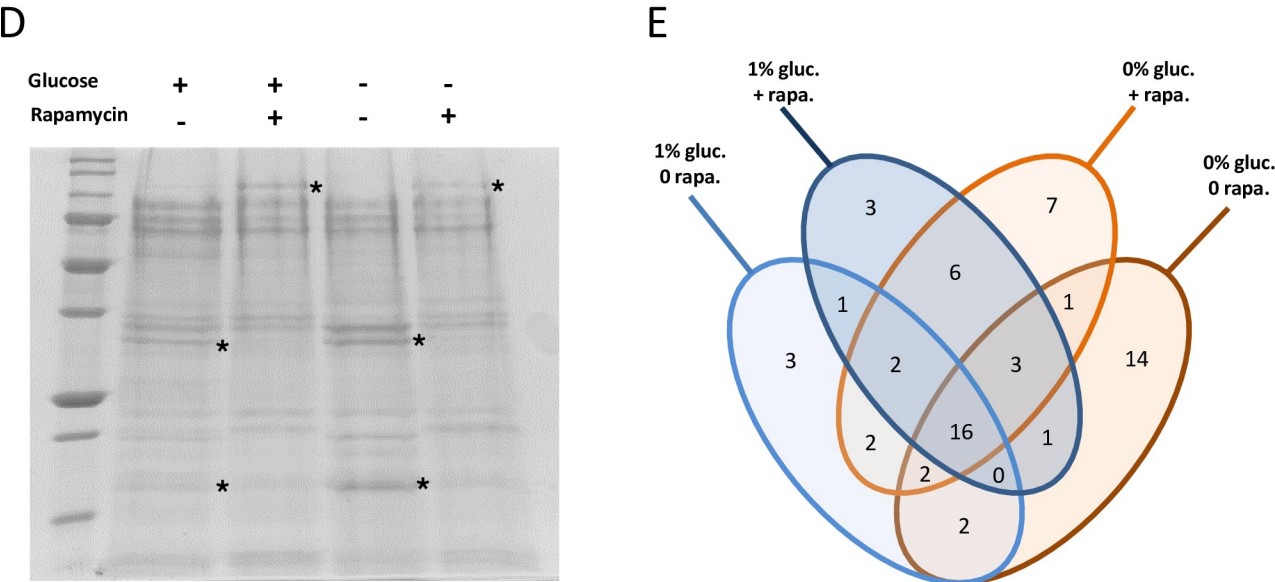

**Fig 2. Rapamycin affects the secretome in *P. chrysosporium*.** Five days old liquid cultures of *P. chrysosporium* were transferred to fresh liquid medium for 48 hours with or without glucose and with or without rapamycin. A. Effect of glucose and rapamycin on dry weight of fungus cultivated in liquid medium. Represented values are means ± SD with n = 6. B. Effect of glucose and rapamycin on the concentration of proteins secreted in the liquid medium as measured using Bradford assays. Values represented are means ± SD with n = 3 C. Beta-glucosidase activity observed in the liquid culture medium containing the secreted enzymes. The activity has been measured using fluorogenic assay. Values represented are means ± SD with n = 6. Two asterisks (**) denote the statistical difference of the added rapamycin medium compare with the control medium (p< 0.05) as determined by the two sample t-test at (A) and (B). Three asterisks (***) and two of them (**) represent (p<0.001) and (p<0.01) as determined by the two sample t-test at (C), respectively. D. Representative Coomassie blue stained SDS-PAGE of the proteins secreted by *P. chrysosporium* in the conditions described above. 13 µg of proteins were loaded for each condition. Asterisk marks highlighted the major change observed (S1 Raw Image). E. Venn diagram representing the number of common and specific proteins found in the four different conditions analyzed. The list of proteins found in each class is provided in S1 Table.

also be found. iii. Proteins identified in point ii are not found in response to rapamycin treatment and iv. more surprisingly we found specific proteins secreted in response to rapamycin and one of them, AGR57_8308, was identified as Fkbp12 (S1 Table).

## Identification of Tor and Fkbp12 in *P. chrysosporium*

In previously studied organisms, rapamycin effect is due to its interaction with two partners: FKBP12 and the FRB domain of TOR [28] (Fig 3B), that leads to the inhibition of the TOR kinase activity. These rapamycin interacting proteins have been searched in the available *P. chrysosporium* strain RP78 genome [23]. A homolog of Fkbp12 was detected by BLASTp using Fkbp12 sequence from *S. cerevisiae* as bait (ScFKBP12, SGDID: S000005079). The identified protein PcFkbp12, coded by the gene AGR57_8308 and located at Scaffold 8:545501–546137, exhibits 63.2% of identity with ScFkbp12.

The second partner binded by Fkbp12-Rapamycin is Tor. The organization of Tor primary sequence is conserved in all eukaryotic organisms, with the presence of several domains from the N terminal part to the C-terminal part: a HEAT repeat domain, a FAT domain, a FRB domain, the Kinase domain and then a FATC domain. The corresponding gene has been found duplicated in fungal lineages (Fig 3A) [16]. Using the sequences from *S. cerevisiae* as bait, two sequences were detected in the genome of *P. chrysosporium* using BLASTp. A first sequence encoded by the gene AGR57_15312 is located on Scaffold 42: 6500–13402 (-). The sequence coding for the FRB domain (PcFRB1) is located between the position 11126 and 11579 of this gene. We found a second hit located on scaffold_20:36724–33343. This sequence returned one transcript sequence AGR57_14232TO. From this gene, the 3' part of the gene, with the sequence of the FRB domain, was missing from the genomic sequence and the transcript. PcFRB domain sequence was amplified by PCR from genomic DNA from *P. chrysosporium* RP78 using the identified sequence coding for PcFRB1 to design primers and subsequently cloned. Sequencing of individual clones revealed the presence of two distinct sequences: the sequence encoded by the gene AGR57_15312 (*PcFRB1*) and another sequence exhibiting 86.8% of identity at nucleotide level and named *PcFRB2*.

Previous transcriptomic data of *P. chrysosporium* were used [29] (NCBI GEO GSM338 1937-39) focusing on the sequencing reads covering the FRB sequence at Scaffold 42. Alignment of those reads revealed the presence of potential SNPs suggesting the presence of two distinct sequences. Using CAP3 sequence assembly program [30], two contigs were obtained, one strictly similar with the sequence coded by AGR57_15312 on scaffold 42 and the other one containing all the mismatches observed in the cloning experiment.

Alignment of FRB sequences from human, *S. cerevisiae*, *S. pombe* and the basidiomycete *P. ostreatus* with PcFRB1 and PcFRB2 from *P. chrysosporium* revealed a high level of conservation (Fig 3). Amino acids required for interaction with rapamycin (W93 and F100) are conserved in those phylogenetically distant species. Residues involved in phosphatidic acid binding in TOR (L24, F31 and Y97) are also conserved. S28, the mutation that is involved in rapamycin resistance in mammalian cells can also be found in all the FRB tested even in both PcFRB (Fig 3A).

The Fkbp12-rapamycin-Tor complexes involving FRB1 or FRB2 from *P. chrysosporium* was generated by homology modeling using a human complex (pdb entry:1fap) followed by an energy minimization (Fig 3B and 3C). The total energy obtained from the YASARA Energy Minimization Server [26] were -32, 174 kcal/mol (score: -0.12) and -32, 257 kcal/mol (score: -0.12) for the refined complexes involving PcFRB1 or PcFRB2, respectively. None of the 4 different residues between PcFRB1 (F29, H71, V86 and K91) and PcFRB2 (Y29, I71, I86 and R91) are involved in complex formation and closed to the rapamycin binding site. Rapamycin is hydrogen bonded to the D38, G54, K55, I57 and Y83 residues from PcFkbp12 (Fig 3D–3F) and also interacts through close contacts with aromatic residues as already described [28]. Among these aromatic residues, W93 and F100 residues from both FRB domains (Fig 3E and

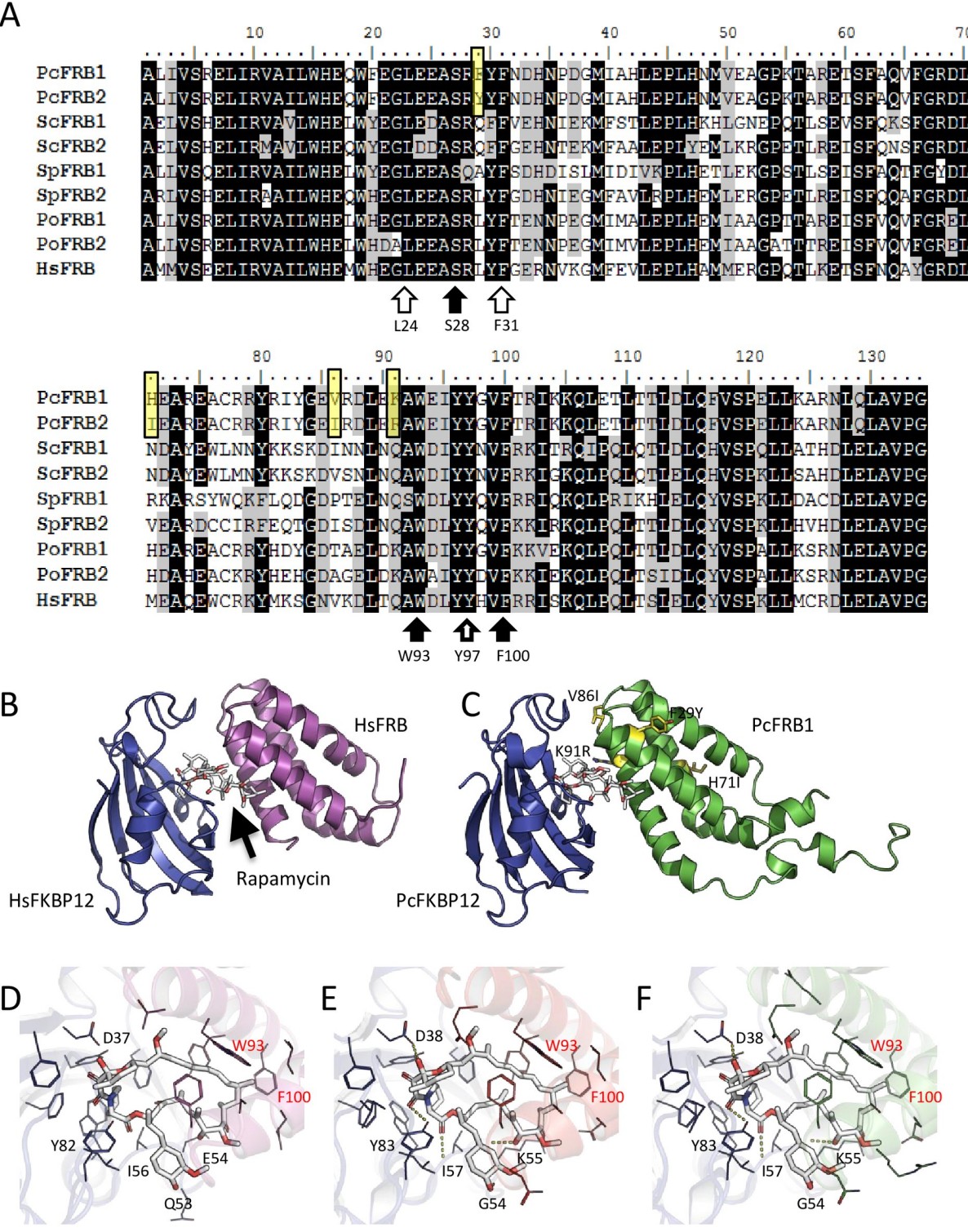

**Fig 3. *P. chrysosporium* possesses two FRB domains that can interact with rapamycin and form TOR_FRB-Rapamycin-FKBP12 complex.**
A. Sequence alignment of the two FRB domains of *P. chrysosporium* with the FRB domain of HsTOR (protein ID: P42345.1), the 2 FRB domains of *S. cerevisiae* (Tor1 SGDID:S000003827 and Tor2: SGDID:S000001686), *S. pombe* (Tor1: SPBC30D10.10c and Tor2: SPBC216.07c from https://www.pombase.org/), *P. ostreatus* (https://mycocosm.jgi.doe.gov/pages/search-for-genes.jsf?organism=PleosPC15_2, TOR ID: 1113153 and TOR2 ID: 1051426). B. Crystal structure of HsFKBP12-HsFRB-rapamycin ternary complex (pdb entry:1fap). C. 3D model of the predicted structure for *P. chrysosporium* FKBP12-rapamycin-FRB complexes. FKBP12, FRB and PcFRB1 are colored in blue, purple and green, respectively. Substitutions occurring between PcFRB1 and PcFRB2 are highlighted as yellow sticks. Secondary structures are shown in

ribbon representation and the rapamycin is shown as white sticks. D-F. Close-up views of the rapamycin binding site in the human FKBP12-rapamycin-FRB complex (D) and in *P. chrysosporium* involving PcFRB1 (E) and PcFRB2 (F). Hydrogen bonded residues to the rapamycin are named in black. W93 and F100 aromatic residues are pointed in red, their positions are given from the beginning of the FRB domain.

3F) are required for interaction with rapamycin. This suggests that rapamycin can interact with the two PcFRB domains.

## Conservation of the TOR pathway in *P. chrysosporium*

In the fungal kingdom, the TOR pathway has been extensively studied in model yeast *S. cerevisiae*, *S. pombe*, and the pathogenic *Candida albicans*. In other fungal lineages, ortholog proteins involved in the TOR pathway have been identified by gene homology and for the *Basidiomycota* lineage, only data for *P. ostreatus* were available presently [16]. The TOR pathway can be divided into three parts, the TOR complexes called TORC1 and TORC2, an upstream part composed of proteins involved in signal transduction and a downstream part including direct targets of the TOR kinase activity. Orthologs identified in *P. chrysosporium* by gene homology for these three parts are summarized in Tables 1, 2 and 3 respectively.

TORC1 is known to control translation at different steps and its activity is sensitive to rapamycin treatment while TORC2 is insensitive to rapamycin and coordinate growth with plasma membrane production and cell wall integrity [8,9,31]. TORC1 and TORC2 have 2 proteins in common: the kinase Tor and a binding partner called Lst8 in *S. cerevisiae* or Wat1 in *S. pombe*. An ortholog, PcLst8, can be found in *P. chrysosporium* (Table 1) with 58% of identity with ScLst8 and 61.5% of identity with SpWat1. TORC1 is also defined by the presence of Kog1. Interacting with Tor, Kog1 recruits targets for TOR kinase activity. The functional homolog of Kog1 in plant and animal cells is RAPTOR [1]. In *P. chrysosporium*, an ortholog called PcKog1 could be identified, this protein exhibiting 30.4% of identity with ScKog1 and 33.3% with SpKog1. Tco89 is TORC1 specific and can be found in yeasts *S. cerevisiae* and *S. pombe*, but could not be identified in *P. chrysosporium*. This is probably due to the low sequence conservation of that protein: SpTco89 and ScTco89 share 11.7% of identity and 19.9% of similarity. In *S. cerevisiae*, in addition to Tor and Lst8, TORC2 is composed of Bit61, Bit2, Avo1, Avo2 and Avo3. Bit61 and Bit2 are paralogs and their sequence share 42.9% of identity and 59.7% of similarity. On the other hand, Bit61 ortholog exists in *S. pombe* but ScBit61 and SpBit61 sequences share 17% of sequence identity and 29.5% of similarity. That low level of sequence homology could explain why those sequences lead to the identification of the same gene (AGR57_12865) in *P. chrysosporium*. For Avo2, which has no known ortholog, no match has been found while searching the *P. chrysosporium* database. PcAvo1 has been identified by its homology with Avo1 but also Sin1 from *S. pombe*. Finally, PcAvo3 could also been retrieved based on its sequence homology with Avo3 and Ste20 from *S. pombe*.

Concerning the upstream part of the TOR pathway, in *S. pombe* and animal cells, TSC1 and TSC2 (Tuberous Sclerosis Complex 1 and 2) form a complex that negatively regulates the GTPase-activity of Rheb (Ras homolog enhanced in brain) to activate TORC1. TSC1 and TSC2 orthologs were identified in all fungal species examined to date with the exception of *S. cerevisiae* [16]. As expected, orthologs named PcTsc1 and PcTsc2 have been detected and also a potential PcRhb1 (Table 2). In *S. cerevisiae*, TORC1 integrates different signals reflecting its regulation by different upstream regulators [32]. On the one hand, TORC1 is regulated by RAG GTPases Gtr1/Gtr2 in complex with Ego1/Ego2/Ego3. This complex is regulated by the GTP exchange factor Vam6. As orthologs for Gtr1, Gtr2 and Vam6 can be found in other fungal lineages [16], we were able to find, in *P. chrysosporium*, genes coding for orthologs of Gtr1,

**Table 1. Identified elements of TOR complexes of *Phanerochaete chrysosporium* in comparison with elements of TOR complexes of *Saccharomyces cerevisiae* and *Schizosaccharomyces pombe*.**

| complex | S. cerevisiae or S. pombe | | | | P. chrysosporium | | | | | |
| | Standard name | SGD Systematic name | SGD ID | Pombase | Proposed name | Gene ID | Location | Protein length | % of identity | % of similarity |
|---|---|---|---|---|---|---|---|---|---|---|
| TORC1/ C2 | Lst8 | YNL006W | S000004951 | | PcLst8 | AGR57_579 | SC001:1196227-1197784(+) | 326 | 58.1 | 74.2 |
| | Wat1/Pop3 | | | SPBC21B10.05c | | | | | 61.5 | 76.5 |
| TORC1 | Kog1 | YHR186C | S000001229 | | PcKog1 | AGR57_11672 | SC013:163355-168458(-) | 1561 | 30.4 | 45.3 |
| | Kog1 | | | SPAC57A7.11 | | | | | 33.3 | 49.5 |
| | Tco89 | YPL180W | S000006101 | SPCC162.12 | Not detected | | | | | |
| TORC2 | Bit2 | YBR270C | S000000474 | | | AGR57_12865 | SC015:519331–521649 (+) | 534 | 13.9 | 22.4 |
| | | | | SPAC6B12.03c | | | | | 12.7 | 21.3 |
| | Bit61 | YJL058C | S000003594 | | | AGR57_12865 | SC015:519331–521649 (+) | 534 | 13.3 | 24.0 |
| | | | | SPCC777.08c | | | | | 14.4 | 24.2 |
| | Avo1 | YOL078W | S000005438 | | PcAvo1 | AGR57_5388 | SC005:594533–598186 (-) | 908 | 19.2 | 31.6 |
| | Sin1 | | | SPAPYUG7.02c | | | | | 22.4 | 37.8 |
| | Avo2 | YMR068W | S000000895 | | Not detected | | | | | |
| | Avo3/Tsc11 | YER093C | S000000895 | | PcAvo3 | AGR57_8271 | SC008:473094-478330(-) | 1334 | 18.6 | 37.1 |
| | Ste20 | | | SPBC12C2.02c | | | | | 30.2 | 50.9 |

*Saccharomyces cerevisae* data from https://www.yeastgenome.org/

*Shizosaccharomyces pombe* data from https://www.pombase.org/

Blast done at http://fungidb.org/fungidb/

% of identity and similarity at https://www.ebi.ac.uk/Tools/psa/emboss_needle/

Gtr2 and Vam6 but not for Ego1 and Ego3. Gtr1 is also regulated by SEACIT (Seh1 –associated subcomplex inhibiting TORC1) which is bound and negatively regulated by SEACAT (Seh1 –associated subcomplex activating TORC1). SEACIT is composed of Npr2, Npr3 and Iml1 and SEACAT of Sec13, Seh1, Sea2, Sea3, SeA4 (Gonzalez and Hall, 2017). Orthologs of those components can be retrieved in *P. chrysosporium* (Table 2). On the other hand, Gtr2 activity is regulated by Lst4/Lst7 but no orthologs have been found by sequence homology. Finally, TORC1 activity is regulated by Snf1 in response to cytosolic glucose starvation and an ortholog exists in *P. chrysospoiurm*.

For the downstream part of the TOR pathway in *S. cerevisiae*, the best known targets of TORC1 are the PP2A phosphatase Sit4, which forms a complex with Tap42 as regulatory sub-unit and Sch9, an AGC kinase with functional homology to the p70S6 Kinase in animal cells [33]. In *P. chrysosporium*, we were able to find orthologs for those proteins (Table 3). This result is in accordance with results previously obtained in *P. ostreatus* but we were also able to find two orthologs for another target of TORC2, Ypk2, while none of them were found in *P. ostreatus* [16]. The same orthologs were found for Ypk1 and interestingly, the two identified orthologs have an even better identity with Gad8, the target of TORC1 and TORC2 in *S. pombe* than with Ykp1 (Table 3). Finally, an ortholog has been identified for the TORC2 target Pkc1. All together, these results suggest that TOR pathway from *P. chrysosporium* is more closely related to *S. pombe* than *S. cerevisiae*.

**Table 2. Identified upstream elements of the TOR signaling pathway of *Phanerochaete chrysosporium* in comparison with upstream elements of TOR pathway of *Saccharomyces cerevisiae* and *Schizosaccharomyces pombe*.**

| complexe | S. cerevisiae or S. pombe | | | | P. chrysosporium | | | | | |
| --- | --- | --- | --- | --- | --- | --- | --- | --- | --- | --- |
| | Standard name | SGD Systematic name | SGD ID | Pombase | Proposed name | Gene ID | Location | Protein length | % of identity | % of similarity |
| Tsc1/Tsc2 | Tsc1 | | | SPAC22F3.13 | PcTsc1 | AGR57_528 | SC001:1108704-1111851(-) | 925 | 20.8 | 36.6 |
| | Tsc2 | | | SPAC630.13c | PcTsc2 | AGR57_9302 | SC008:783034-788376(+) | 1763 | 18.8 | 32.1 |
| | Rhb1 | YCR027C | S000000622 | | PcRhb1 | AGR57_1568 | SC002:138810–139648(+) | 194 | 40.1 | 63.2 |
| | | | | SPBC428.16c | | | | | 56.6 | 71.4 |
| Gtr1/Gtr2 | Gtr1 | YML121W | S000004590 | | PcGtr1 | AGR57_793 | SC001:1652706-1653971(+) | 348 | 40.7 | 57.3 |
| | | | | SPBC337.13c | | | | | 45.6 | 64.8 |
| | Gtr2 | YGR163W | S000003395 | | PcGtr2 | AGR57_4381 | SC004:781482-783110(+) | 435 | 25.7 | 43.7 |
| | | | | SPCC777.05 | | | | | 31.4 | 44.6 |
| | Ego1/Meh1 | YKR007W | S000001715 | | Not detected | | | | | |
| | Ego3/Slm4 | YBR077C | S000000281 | | Not detected | | | | | |
| | Vam6 | YDL077C | S000002235 | | PcVam6 | AGR57_7116 | SC007:84318-88231(-) | 1038 | 17.3 | 30.0 |
| | | | | SPAC23H4.14 | | | | | 23.3 | |
| SEACIT | Npr2 | YEL062W | S000000788 | | PcNpr2 | AGR57_425 | SC001:882467-884614(-) | 632 | 18.2 | 29.8 |
| | | | | SPAC23H3.03c | | | | | 23.2 | 38.1 |
| | Npr3/Rmd11 | YHL023C | S000001015 | | PcNpr3 | AGR57_4386 | SC004:792751-795311(-) | 781 | 16.5 | 29.4 |
| | | | | SPBC543.04 | | | | | 20.5 | 36.8 |
| | | YJR138W | S000003899 | | PcIml1 | AGR57_11382 | SC012:775902-781246(-) | 1639 | 24.0 | 41.7 |
| | | | | SPBC26H8.04c | | | | | 26.4 | 42.1 |
| SEACAT | Seh1 | YGL100W | S000003068 | | PcSeh1 | AGR57_5195 | SC005:129083-130804(+) | 430 | 30.5 | 45.4 |
| | | | | SPAC15F9.02 | | | | | 29.6 | 47.0 |
| | Sec13/Anu3 | YLR208W | S000004198 | | PcSec13 | AGR57_14171 | SC019:388062-389213(+) | 332 | 51.6 | 64.8 |
| | | | | SPBC215.15 | | | | | 53.6 | 68.3 |
| | Sea2/Rtc1 | YOL138C | S000005498 | | PcSea2 | AGR57_7036 | SC006:1979960-1984607(-) | 1299 | 16.4 | 28.4 |
| | | | | SPAC4F8.11 | | | | | 19.2 | 30.9 |
| | Sea4 | YBL104C | S000000200 | | PcSea4 | AGR57_6153 | SC006:62362-66094(-) | 1123 | 22.9 | 36.6 |
| | | | | SPAC12G12.01c | | | | | 24.4 | 38.2 |
| | Sea3/Mtc5 | YDR128W | S000002535 | | PcSea3 | AGR57_731 | SC001:1518472-1523186(+) | 1238 | 26.0 | 39.9 |
| | | | | SPAC11E3.05 | | | | | 23.8 | 37.4 |
| Lst4/Lst7 | Lst4 | YKL176C/ | S000001659 | SPAC30C2.07 | Not detected | | | | | |
| | Lst7/Bhd1 | YGR057C/ | S000003289 | SPBC24C6.08c | Not detected | | | | | |
| | Snf1 | YDR477W | S000002885 | | PcSnf1 | AGR57_13796 | SC018:113339–115564 (+) | 607 | 36.8 | 50.8 |
| | Ssp2 | | | | | | | | 35.4 | 49.2 |

*Saccharomyces cerevisiae* data from https://www.yeastgenome.org/

*Shizosaccharomyces pombe* data from https://www.pombase.org/

Blast done at http://fungidb.org/fungidb/

% of identity and similarity at https://www.ebi.ac.uk/Tools/psa/emboss_needle/

**Table 3. Identified downstream elements of the TOR signaling pathway of *Phanerochaete chrysosporium* in comparison with downstream elements of TOR pathway of *Saccharomyces cerevisiae* and *Schizosaccharomyces pombe*.**

| Target of | S. cerevisiae or S. pombe | | | | P. chrysosporium | | | | | |
|---|---|---|---|---|---|---|---|---|---|---|
| | Standard name | SGD Systematic name | SGD ID | Pombase | Proposed name | Gene ID | Location | Protein length | % of identity | % of similarity |
| TORC1 | **Tap42** | YMR028W | S000004630 | | PcTap42 | AGR57_809 | SC001:1683869–1685426 (-) | 389 | 24.1 | 42.7 |
| | | | | SPCC63.05 | | | | | 27.1 | 43.0 |
| TORC1 | **Sit4** | YDL047W | S000002205 | | PcSit4 | AGR57_15090 | SC027:124552–125781 (+) | 305 | 55.9 | 75.9 |
| | | | | SPCC1739.12 | | | | | 58.2 | 79.1 |
| TORC1 | **Sch9** | YHR205W | S000001248 | | PcSch9 | AGR57_2503 | SC002:2055146–2058000 (+) | 675 | 34.6 | 46.0 |
| | | | | SPAC1B9.02c | | | | | 36.3 | 47.9 |
| TORC2 | **Ypk1** | YKL126W | S000001609 | | PcYpk1-2 or PcGad8 | AGR57_11425 | SC012:862786-864920(+) | | 44.3 | 55.9 |
| | | | | | | AGR57_8253 | SC008:439512-441528(+) | | 56.6 | 70.3 |
| TORC2 | **Ypk2** | YMR104C | S000004710 | | PcYpk1-2 or PcGad8 | AGR57_11425 | SC012:862786-864920(+) | 547 | 41.9 | 53.7 |
| | | | | | | AGR57_8253 | SC008:439512-441528(+) | 549 | 39.9 | 53.8 |
| TORC1/2 | **Gad8** | | | SPCC24B10.07 | PcYpk1-2 or PcGad8 | AGR57_8253 | SC008:439512-441528(+) | 549 | 51.7 | 64.0 |
| | | | | | | AGR57_11425 | SC012:862786-864920(+) | 547 | 53.6 | 68.3 |
| TORC2 | **Pkc1** | YBL105C | S000000201 | | | AGR57_14288 | SC020: 136736–140671 (+) | | 36.8 | 51.5 |

*Saccharomyces cerevisiae* data from https://www.yeastgenome.org/

*Shizosaccharomyces pombe* data from https://www.pombase.org/

Blast done at http://fungidb.org/fungidb/

% of identity and similarity at https://www.ebi.ac.uk/Tools/psa/emboss_needle/

## Conclusion

With this study, we found that the TOR pathway play a central role in *P. chrysosporium*. Using rapamycin, we show that the TOR pathway is involved in conidia germination, vegetative growth and the secreted protein composition. Interestingly, rapamycin treatment leads to the identification of proteins secreted in response to rapamycin. One of them is PcFKBP12 which is an unusual location for this protein. In *A. fumigatus*, change in localization of AfFKBP12 has been observed in response to the immunosuppressant FK506 but not its secretion [34]. Due to amino acid conservation, the FRB domain of TOR and FKBP12 from *P. chrysosporium* can interact with rapamycin. The other components of the TOR pathway in *P. chrysosporium* have been identified and more components are related to the TOR pathway from *S. pombe* relative to *S. cerevisiae*.

## Supporting information

**S1 Movie. The effect of rapamycin on *P. chrysosporium* growth during a diffusion test.** (AVI)

**S1 Table. Table of data on the proteins identified during proteomic experiment.** (XLSX)

**S1 Raw Image. Unmodified picture of the Coomassie blue stained SDS-PAGE SDS PAGE presented in Fig 2D.**
(PDF)

## Acknowledgments

We thank Dr. JB Vincourt from the proteomics core facility of UMS2008 IBSLor UL-CNRS-INSERM for performing mass spectrometric analysis.

We thank Professor Jean-Pierre Jacquot for his help and comments during manuscript preparation.

## Author Contributions

**Conceptualization:** Rodnay Sormani.

**Investigation:** Duy Vuong Nguyen, Thomas Roret, Antonio Fernandez-Gonzalez, Annegret Kohler, Rodnay Sormani.

**Supervision:** Mélanie Morel-Rouhier, Eric Gelhaye.

**Writing – original draft:** Duy Vuong Nguyen, Rodnay Sormani.

**Writing – review & editing:** Duy Vuong Nguyen, Thomas Roret, Antonio Fernandez-Gonzalez, Annegret Kohler, Mélanie Morel-Rouhier, Eric Gelhaye, Rodnay Sormani.

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
