## [Decision Letter · Decision Letter 0]

26 Nov 2019

PONE-D-19-29130

Target Of Rapamycin pathway in the white-rot fungus Phanerochaete chrysosporium

PLOS ONE

Dear Dr sormani,

Thank you for submitting your manuscript to PLOS ONE. After careful consideration, we feel that it has merit but does not fully meet PLOS ONE’s publication criteria as it currently stands. Therefore, we invite you to submit a revised version of the manuscript that addresses the points raised during the review process.

We were able to secure only one review. Accordingly, the decision is based on that reviewer's comments and my own reading of the manuscript. I generally concur with the reviewer's points, particularly the clarity of the proteomic methods and analysis.  Please consider revising these sections and, in addition to the reviewer's suggestions, clearly describe the experimental details. For example, precisely how many replicates were processed? I understand that the data may not lend itself for quantitative (statistical) analyses, but an explanation of the scoring (Table 2S) would be helpful to many readers. Consider footnotes to the supplemental table. 

In addition to the reviewer, please also consider these minor suggestions:

Line 54. Replace with ref #17

Line 150. This is confusing.  What "rap mutants"?

Line 168. "fumigate"?

Line 227. Consider colored highlighting of AGR57_3808 in table S2.

Line 334. Consider "more closely related to"

Line 354. Consider "S. pombe relative to S. cerevisiae"

We would appreciate receiving your revised manuscript by Jan 10 2020 11:59PM. To enhance the reproducibility of your results, we recommend that if applicable you deposit your laboratory protocols in protocols.io, where a protocol can be assigned its own identifier (DOI) such that it can be cited independently in the future. For instructions see: http://journals.plos.org/plosone/s/submission-guidelines#loc-laboratory-protocols

Also, please note the journal's requirement  regarding blot/gel data: PLOS ONE now requires that submissions reporting blots or gels include original, uncropped blot/gel image data as a supplement or in a public repository. This is in addition to complying with our image preparation guidelines described at https://journals.plos.org/plosone/s/figures#loc-blot-and-gel-reporting-requirements. These requirements apply both to the main figures and to cropped blot/gel images included in Supporting Information. If the manuscript is positively reviewed, we will ask the authors to provide any missing raw image data for blot/gel results when they submit their first revision. As part of your review, please ensure that figures reporting blot or gel images comply with the journal's image preparation guidelines and that the original data are provided following the journal's request.  If you have any questions or concerns about blot/gel figures or data for this submission, please email us at plosone@plos.org before issuing a decision letter.

We look forward to receiving your revised manuscript.

Kind regards,

Daniel Cullen

Academic Editor

PLOS ONE

Journal Requirements:

Reviewers' comments:

Reviewer's Responses to Questions

**Comments to the Author**

1. Is the manuscript technically sound, and do the data support the conclusions?

Reviewer #1: Partly

2. Has the statistical analysis been performed appropriately and rigorously? 

Reviewer #1: No

3. Have the authors made all data underlying the findings in their manuscript fully available?

Reviewer #1: No

4. Is the manuscript presented in an intelligible fashion and written in standard English?

Reviewer #1: Yes

5. Review Comments to the Author

Reviewer #1: In this manuscript the authors investigate the conservation of the TOR signaling network in the white-rot fungus Phanerochaete chrysosporium. They observe that rapamycin blocks conidia germination and hyphal growth. They also provide preliminary data to suggest that the composition of proteins secreted by this organism is altered by carbon source and rapamycin.

Main concern:

The proteomics experiments are not well described. How quantitative are these results? What is the FDR rate? Said another way, how can one be sure that proteins observed in one condition are really only present in that condition? This point needs to be readdressed prior to acceptance.

Specific comments

i) line 183: Is the exponential phase really 30.9 hours? There seems to be a shift at about 25hrs. How is exponential phase defined?

ii) Paragraph starting on line 211: Until line 217, I do not see how the specific numbers cited correspond to the Venn diagram in Figure 2E. Shouldn’t a main point here be that 16 proteins are secreted only in the presence of rapamycin and/or 17 are secreted only in the absence of rapamycin?

iii) Why does Table S2 list sucrose as the carbon source?

iv) Line 221 ACR57 should be AGR57

v) Paragraph starting on line 284 – a more current review of TORC2 functions can be found here: https://doi.org/10.3390/biom7030066

vi) Reference 16 is now outdated.

Use this for ortholog searches

https://www.cell.com/cell-metabolism/pdfExtended/S1550-4131(17)30422-9

From line 292 on the paragraph needs to be rewritten for comprehensibility. Note that Tco89 and Bit61 orthologs exist in S. pombe; Ego1 and Ego3 are conserved in metazoans but are difficult to detect in sequence searches.

What about orthologs of SEACIT, SEACAT, Lst4/7? Tip41, Ypk1 and Ypk3?

vii) Legend for figure 3 is not clearly written. Line 495 does not accurately describe figure which has structures from other organisms. Line 502 – where are W89 and F96 in the figure? Why are the red numbers in 3D so discordant with the red numbers in E and F? The species origins for each protein domain should be labelled directly in the figure.

6. PLOS authors have the option to publish the peer review history of their article (what does this mean?). If published, this will include your full peer review and any attached files.

Reviewer #1: No

---

## [Author Response · Author response to Decision Letter 0]

5 Jan 2020

Dear Pr Cullen,

Dear Reviewers,

You will find enclosed my revised version of our manuscript “Target Of Rapamycin pathway in the white-rot fungus Phanerochaete chrysosporium” PONE-D-19-29130. Your advices have been useful and i wish you will find this version improved compare to the previous one.

Before reading the manuscript, let me answering in this letter to the different points that you raised:

“….We were able to secure only one review. Accordingly, the decision is based on that reviewer's comments and my own reading of the manuscript. I generally concur with the reviewer's points, particularly the clarity of the proteomic methods and analysis. Please consider revising these sections and, in addition to the reviewer's suggestions, clearly describe the experimental details. For example, precisely how many replicates were processed? I understand that the data may not lend itself for quantitative (statistical) analyses, but an explanation of the scoring (Table 2S) would be helpful to many readers. Consider footnotes to the supplemental table.”

Description of the experimental procedure has been modified to explain that 6 biological repeats have been used to prepare the samples, this information is repeated in the result part.

Footnotes explaining scoring have been added at the end of the supplemental table2.

Explanation of the significance has been added at the end of mat & meth (those sentences have been written under the supervision of Dr JB Vincourt which is the service provider and thanks in Acknowledgements section).

“In addition to the reviewer, please also consider these minor suggestions:

Line 54. Replace with ref #17”

Done

“Line 150. This is confusing. What "rap mutants"?”

That is obviously a copy/ paste error and a default in our reviewing process. There is no rap mutant in this manuscript. 

“Line 168. "fumigate"?”

fumigatus, changed has been done.

“Line 227. Consider colored highlighting of AGR57_3808 in table S2.”

Done

“Line 334. Consider "more closely related to"”

Done

“Line 354. Consider "S. pombe relative to S. cerevisiae"”

Done

“5. Review Comments to the Author

Main concern:

The proteomics experiments are not well described. How quantitative are these results? What is the FDR rate? Said another way, how can one be sure that proteins observed in one condition are really only present in that condition? This point needs to be readdressed prior to acceptance.”

To answer to your questions several modifications have been done:

Description of the experimental procedure has been modified. Number of biological repeat has been added, significance of the scoring has been describe as follow: 

“The minimal mascot score for peptides was set to 20 and that for proteins was set to 80. Results were cross-validated by interrogating an irrelevant database using the same criteria. Proteins were considered found in a given sample only if identified with a score above 80. When confronted to the random decoy strategy, this score resulted in a false discovery rate < 1%. The other way around, proteins were considered missing only if they were not identified at all. We could not calculate the effective lack of discovery rate in this specific experiment, but in other, technically similar contexts, such proteins exhibited levels at least 20-fold lower than those identified with a score of 80, with less than 2% errors.”

Information has been added in result part and footnotes have been added at the end of the supplemental data.

“Specific comments

i) line 183: Is the exponential phase really 30.9 hours? There seems to be a shift at about 25hrs. How is exponential phase defined?”

We determine the beginning of the stationary phase when no more growth is observed, that point also determine the end of the exponential phase. 

“ii) Paragraph starting on line 211: Until line 217, I do not see how the specific numbers cited correspond to the Venn diagram in Figure 2E. Shouldn’t a main point here be that 16 proteins are secreted only in the presence of rapamycin and/or 17 are secreted only in the absence of rapamycin?”

This paragraph has been rewritten.

“iii) Why does Table S2 list sucrose as the carbon source?”

Corrected, it was glucose and not sucrose.

“iv) Line 221 ACR57 should be AGR57”

Done

“v) Paragraph starting on line 284 – a more current review of TORC2 functions can be found here: https://doi.org/10.3390/biom7030066

vi) Reference 16 is now outdated.

Use this for ortholog searches

https://www.cell.com/cell-metabolism/pdfExtended/S1550-4131(17)30422-9

From line 292 on the paragraph needs to be rewritten for comprehensibility. Note that Tco89 and Bit61 orthologs exist in S. pombe; Ego1 and Ego3 are conserved in metazoans but are difficult to detect in sequence searches.

What about orthologs of SEACIT, SEACAT, Lst4/7? Tip41, Ypk1 and Ypk3?”

Points (v) and (vi) lead us to modify tables 1,2 and 3 which have been enlarged.

Corresponding paragraphs have been rewritten in accordance to those new results.

“vii) Legend for figure 3 is not clearly written. Line 495 does not accurately describe figure which has structures from other organisms. Line 502 – where are W89 and F96 in the figure? Why are the red numbers in 3D so discordant with the red numbers in E and F? The species origins for each protein domain should be labelled directly in the figure.”

Legend has been rewritten in accordance to reviewer comments.

Figure 3 has been modified according to the comments.

W89 and F96 were W93 and F100 according to Fig. 3A, this has been corrected.

Numbers in 3D were given according to HsTOR sequence and in 3E according to PcFRB sequence. Numbers have been changed and homogenized with Fig. 3A. This is indicated in the legend.

I wish you will agree with the revisions we have done,

Best regards,

Rodnay Sormani

---

## [Editor Report · Decision Letter 1]

29 Jan 2020

Target Of Rapamycin pathway in the white-rot fungus Phanerochaete chrysosporium

PONE-D-19-29130R1

Dear Dr. sormani,

We are pleased to inform you that your manuscript has been judged scientifically suitable for publication and will be formally accepted for publication once it complies with all outstanding technical requirements.

With kind regards,

Daniel Cullen

Academic Editor

PLOS ONE
---

## [Editor Report · Acceptance letter]

10 Feb 2020

PONE-D-19-29130R1 

Target Of Rapamycin pathway in the white-rot fungus *Phanerochaete chrysosporium*

Dear Dr. Sormani:

I am pleased to inform you that your manuscript has been deemed suitable for publication in PLOS ONE. Congratulations! Your manuscript is now with our production department. 

With kind regards,

on behalf of

Dr. Daniel Cullen 

Academic Editor

PLOS ONE